# Prevalence, knowledge and practices on prevention and management of overweight and obesity among adults in Dodoma City, Tanzania

**Safiness Simon Msollo** [1] *, **Gosbert Lukenku Shausi**[2], **Akwilina Wendelin Mwanri**[1]

1 Department of Human Nutrition and Consumer Sciences, Sokoine University of Agriculture in Morogoro, Morogoro, Tanzania, 2 Department of Agricultural Extension and Community Development, Sokoine University of Agriculture in Morogoro, Morogoro, Tanzania

* msollosafiness@yahoo.co.uk

**Data Availability Statement:** All relevant data are within the paper and its Supporting Information files.

## Abstract

### Background

Overweight/obesity is increasing leading to high rates of non-communicable diseases. The study aimed to assess the prevalence, knowledge and practices on prevention and management of overweight/obesity among adults in Dodoma region.

### Methodology

A cross- sectional research was conducted among 313 randomly selected adults aged 25–65 years. Face to face interviews were conducted to obtain demographic information, knowledge on preventing and managing overweight/obesity using a pre-tested questionnaire. Weight and height were measured following standard procedures and nutrition status was categorized using WHO criteria. Dietary intake was assessed using qualitative 24 hours recall to obtain dietary diversity score. Data was analyzed using the SPSS™ Version 20 to obtain descriptive and inferential statistics.

### Results

About 62.6% (n = 196) of the participants were females. The overall prevalence of overweight/obesity was 59.7% (n = 186) of which 28% (n = 88) were overweight and 31.3% (n = 98) obese. Obesity was three times higher among females (41.8% vs 13.7%) than males. Overweight/obesity was positively associated with female sex (Adjusted OR 2.34; 95%CI: 1.235–4.68; p = 0.001), low knowledge (Adjusted OR 2.15; 95%CI: 1.22–3.81; p = 0.008) and negatively associated with dietary diversity score (Adjusted OR 0. 452; 95%CI: 0.199–1.87; p = 0.03). About 76% of respondents reported that overweight/obesity is a result of consuming high energy (38.8%; n = 92) and high fat foods (27%; n = 63). More than half of the respondents reported to be insensitive with kind of foods they consume and for those who were sensitive, 64% do so to avoid diseases. Furthermore, 60% control their weight by

**Funding:** This study was supported by the SUARIS-One project from Sokoine University of Agriculture in Morogoro, Tanzania (DPRTC/R/126/COA/5/202). The funders had no role in study design, data collection and analysis, decision to publish, or preparation of the manuscript.

**Competing interests:** The authors have declared that no competing interests exist.

doing physical exercises while 18% by both physical exercises and diet management. However, about 43% did not know foods exposing individuals to health problems.

## Conclusion and recommendations

High prevalence of overweight/obesity was observed and significantly associated with female sex, low dietary diversity and knowledge on overweight/obesity prevention. This creates a need to improve knowledge on prevention of overweight/obesity.

## Introduction

Overweight and obesity are states of abnormal or excessive fat deposition in the body to the extent that they impair health of an individual hence, constituting an important public health problem [1]. Obesity is the term used to indicate the high range of weight for an individual's given height that is associated with adverse health effects [2]. For an adult's population, overweight and obesity are based on set cutoff points which are directly related to an individual's body mass index (BMI) calculated from weight in kilograms divided by the square of height in meters. Prevalence of overweight and obesity are increasing among nations all over the world. The global prevalence of overweight/obesity (body mass index [BMI] $\geq 25$ kg/m$^2$) for overweight and obesity (BMI $\geq 30$ kg/m$^2$) among adults aged $\geq$18 years were 39% and 13%, in 2016 respectively [3].

Overweight and obesity are complex conditions of multifactorial origin, which result from interplay between heterogenic factors, rising from an individual's eating behaviors, physical activity and energy expenditure [4]. Obesity is basically influenced by a combination of biological, genetic, social, environmental, and behavioral factors [5]. Both lifestyle and environmental factors act in a synergistic manner to increase the rate of obesity in developing countries [6].

Overweight and obesity are major determinants of a number of non-communicable diseases (NCDs) including cardiovascular and kidney diseases, type 2 diabetes mellitus, some cancers, musculoskeletal disorders, and other chronic diseases [7, 8].

Tanzania is not excluded as it has high rates of overweight and obesity which are reported to rise from 28% in 2015 to 31.5% in 2018 among women of reproductive age and 29.4% in Dodoma region specifically [9, 10]. Also, the prevalence of abdominal obesity was significantly higher among urban dwellers (33.56%) than their rural counterparts in the region [11]. The reported high rates of overweight and obesity in the area need attention as the problem might be alarming in the near future with the increase rate of NCDs.

Nutrition knowledge plays a significant role in advancing healthy eating habits by ensuring that essential nutrient requirements are met to prevent malnutrition [12]. People who are aware of the relationships between specific health conditions and poor nutrition are better positioned to watch and manage their weight through their diet choices. Kolodinsky et al. [13] pointed out that, knowledge of dietary guidelines and healthy eating habits among adults are positively correlated. A change in eating behaviour starts from the awareness of the significance of healthy eating behaviour and knowledge of the types of food to eat. However, the relationship between what people do and what they know is weak. This is because knowledge parse does not automatically produce a behavioral change in humans but can act as a powerful instrument when people decide to change [14].

Although high prevalence of overweight and obesity are reported in the country, Dodoma in particular, there is paucity of data about knowledge and practices on prevention and

management of the conditions. Therefore, this study aimed to establish the knowledge gaps on the prevention and management for overweight and obesity among adults as an evidence for designing an intervention.

## Materials and methods

### Description of the study area, population and study design

Dodoma city is one among the seven districts in Dodoma region. The City is bordered by Chamwino district to the East and Bahi district to the West. It lies between Latitudes 6.000 and 6.300 South, and Longitude 35.300 and 36.020 East. It covers an area of 2,769 square km, which is characterized with both urban and rural qualities. Administratively, it is divided into 4 divisions, 41 wards, 18 villages, 170 streets and 89 hamlets. According to the population and housing census of 2012, Dodoma District had 410,956 people of which 196,487 are males and 211,469 females with the estimated annual population growth rate of 2.7% [15].

A cross-sectional research with quantitative approach was conducted as part of NCDs study among male and female adults between the ages of 25–65 years in urban areas of Dodoma region. Dodoma region was selected due to its fast growth as a result of shifting of the Ministries. The study excluded pregnant women and those who were on-transit.

### Sampling techniques

Multistage sampling technique was employed to obtain the streets to be included in the study. In this technique, purposive sampling was applied to obtain one district located in urban area in Dodoma region. Thereafter, simple random sampling was used to select 4 wards from the selected district and one street from each of the selected wards. From each selected street, a list of households was provided and again a simple random selection using a table of random numbers was applied to obtain a total of 313 households for the four streets while considering sampling proportional to number of households in each street. In this case 50% of the study households were selected from Mwaja Street in Chamwino ward, 26% from Makole Street in Makole ward, 16% form Kikuyu Street in Kikuyu West ward and 8% from Tofiki Street in Viwandani ward. The selection of household representatives was done randomly in the household with more than one adult who met the selection criteria after stratifying them according to sex.

### Sample size determination

Eligible adults were selected from each participating site until a total of 313 adults were recruited for participation. This sample size was obtained using the formula for prevalence studies [16]:

$$n = \frac{z^2 * p * q}{d^2}$$

Where: n = desired sample size
Z = standard normal deviation set at 1.96 corresponding to 95% Confidence Interval [CI]
p = proportion of the target population with DR-NCDs
q = 1.0 − p
d = degree of accuracy desired (0.05)
The national prevalence of hypertension (26%) [17] was used to represent the prevalence of DR-NCDs with an assumed response rate of 95% and a non-respondent rate of 5%. The

prevalence of hypertension was used during sample calculation because the study was part of the big study on DR-NCDs.

**Training of enumerators and pretesting of research tools.** Before the actual data collection processes enumerators who were nutritionists with enough experience on human research, were trained on the study protocol and how to use the research tools. The training was done for three days using questionnaire translated into Kiswahili that is understood by the study participants. After the training, all trainees were involved in pre-testing the tools with 20 randomly selected adults in urban Morogoro. The pre-test results were discussed, and appropriate changes were made to improve the research tools.

## Data collection procedure

The study was introduced to participants who met the selection criteria. Information sheet was read to the selected participants who were requested to sign an informed consent form for participation after understanding and agreeing with the aim and procedure of the study. After consenting, participants were subjected to an interview which was conducted in their households and were given some identification tags which they presented at the center for anthropometric measurements to ensure that the same person who was interviewed is involved in measurements.

## Demographic characteristics

Demographic characteristics, smoking and alcohol intake habits were assessed using a questionnaire with close and open-ended questions adapted from STEPS instrument for NCDs [18] and modified to fit the study's context.

## Anthropometric measurements

Weight was measured in kilogram (kg) with minimum clothing and without shoes by using a digital bathroom weighing scale (SECA®)-Germany), placed on a flat surface and recorded to the nearest 0.1kg. A minimum of two readings were recorded and an average been calculated.

Height was measured using a portable stadiometer (SECA®—Germany) where the participants were requested to stand up straight against the backboard with the body weight evenly distributed and both feet flat on the platform, with the heels placed together and toes apart. The back of the head, shoulder blades, buttocks and heels were made to contact with the backboard. The head was aligned in the Frankfort horizontal plane and the stadiometer head piece been lowered to rest firmly on top of the participant's head, with sufficient pressure to compress the hair. The measurements were taken in duplicate and recorded at the nearest 0.1 cm. This height, together with weight, was used to calculate BMI of the adults in kilogram per meter squares. Subject with BMI $< 18.5$ kg/m$^2$ was classified as underweight, 18.5–24.9 kg/m$^2$ as normal, 25–29.9 kg/m$^2$ overweight and $\geq 30$ kg/m$^2$ as obese [19].

## Assessment of dietary intake

Dietary intake was assessed using an individual dietary diversity score whereby a 24 hours recall was used to assess the foods consumed during breakfast, lunch, dinner and anything eaten between meals (snacks). The assessed foods were grouped into 16 food groups to obtain their dietary diversity score (DDS). In this case, 'yes' was assigned for anything eaten within 24 hours with the amount $\geq 15$g and 'no' for anything consumed in $<15$g or not consumed at all (Table 1).

**Table 1. Dietary Diversity Score (DDS).**

| | Food groups | Example of foods | Yes = 1/ No = 0 |
|---|---|---|---|
| 1 | Cereals | corn/maize, rice, wheat, sorghum, millet or any their products (e.g. bread, noodles, porridge, ugali, porridge or paste | |
| 2 | White roots and tubers | white potatoes, white yam, white cassava, or other foods made from roots | |
| 3 | Vitamin A rich vegetables and tubers | pumpkin, carrot, squash, or sweet potato that are orange inside | |
| 4 | Dark green leafy vegetables | wild forms + locally available vitamin A rich leaves such as amaranth, cassava leaves, kale, spinach | |
| 5 | Other vegetables | tomato, onion, eggplant other locally available vegetables | |
| 6 | Vitamin A rich fruits | ripe mango, cantaloupe, ripe papaya, dried peach, and 100% fruit juice made from these + other locally available vitamin A rich fruits | |
| 7 | Other fruits | wild fruits and 100% fruit juice made from these | |
| 8 | Organ meat | liver, kidney, heart or other organ meats or blood-based foods | |
| 9 | Flesh meat | beef, pork, lamb, goat, rabbit, game, chicken, duck, other birds, insects | |
| 10 | Eggs | eggs from chicken, duck, or any other egg | |
| 11 | Fish and sea foods | fresh or dried fish or shellfish | |
| 12 | Legumes and nuts | dried beans, dried peas, lentils, nuts, seeds or foods made from these (eg. peanut butter) | |
| 13 | Milk and milk products | milk, cheese, yogurt or other milk products | |
| 14 | Oils and fats | oil, fats or butter added to food or used for cooking | |
| 15 | Sweets | sugar, honey, sweetened soda or sweetened juice drinks, sugary foods such as chocolates, candies, cookies and cakes | |
| 16 | Spices, condiments and beverages | spices (black pepper, salt), condiments (soy sauce, hot sauce), coffee, tea, alcoholic beverages | |
| | | Did you eat anything (meal or snack) OUTSIDE the home yesterday? | |

**Source**: Food and Agriculture Organization [21]

For most age-groups there is no agreement on a cut-off to define a dichotomous indicator therefore it is recommended to use the score as a continuous indicator to calculate mean and median [20].

## Assessment of knowledge and practices on prevention and management of overweight and obesity among adults

Face to face interviews were conducted among adults using a pre-tested questionnaire to get their level of knowledge on prevention of overweight and obesity. Multiple answers were allowed in relevant questions and coded afterwards during data management. Information collected included but not limited to respondents' views on the causes, consequences, and risk factors, prevention and management of overweight and obesity, common practices used to prevent, screen and manage weight were also assessed using Knowledge Attitude and Practice (KAP) questionnaire, which included both close and open-ended questions adapted from the Tanzania STEPS survey [18].

The interpretation of knowledge was done based on the scores obtained, whereby those participants who replied correctly were interpreted as having adequate knowledge and graded with a score of one (1) and those with incorrect answers were graded a score of zero (0) for each question which were then converted into percentages. Interpretation of knowledge was done according to Dhyani et al. [22] where excellent knowledge was regarded as $\geq 75\%$

correct, good knowledge 50%≤X<75% correct; average 25%≤X<50% as well as poor knowledge as < 25%.

## Data analysis

Data was cleaned, coded, entered and analyzed using the Statistical Package for Social Science™ (SPSS™) Version 20. In this software descriptive statistics such as frequencies, means, median and percentages were obtained for demographic information, knowledge and prevalence for describing nutrition status of adults. Associations among factors were obtained by binary logistic regression analysis (Univariate and multivariate). The outcomes, which are knowledge on prevention of overweight and obesity, were dichotomized into either having knowledge or not having knowledge and being overweight/obesity or being normal. In this case a multiple logistic regression analysis was used to find associations of different factors with overweight and obesity using stepwise backward elimination [23]. Crude and adjusted odd ratios were obtained for each factor associated with overweight and obesity at p < 0.05.

## Results

### Demographic characteristics of the participants

About 71% of the respondents were adults of a reproductive age (25–49) with mean age of 41 ±12.5 years where more than half (62.6%) were female participants. Among the participated adults, 45% were formally married and 53.7% completed primary school. Majority of the households (87.9%) had 1–4 adults (>18 years old) living together with the average household size of (adults >18 years) approximately 3±1.7 people (Table 1). More than half of the participants (66.8%) were self-employed largely in small business (54.6%) (Table 2).

### Nutrition status among adults based on BMI

The overall prevalence of overweight/obesity was about 60% (n = 186) of which 28% (n = 88) of the participants were overweight and 31.3% (n = 98) were obese. Prevalence of underweight was very low (2.2%) (Fig 1).

The mean BMI of the participants was 27.33kg/m$^2$ ± 6.4. Overweight (28.6% vs 27.4%) and obesity (41.8% vs 13.7%) were found to be significantly higher among female than male participants (p<0.001) (Table 3).

### Commonly consumed food groups

The results show that the mean DDS for the study population was 4.7 food groups where the most consumed food groups in 24 hours were oil and fats which were consumed by all cereals 265(84.8%), Vitamin A rich vegetables and tubers 208 (66.5%) followed by spices, condiments and beverages (185(59.1%) of which salt and soft drinks were highly consumed in this group (Table 4).

### Factors associated with overweight/obesity

Overweight/obesity was positively associated with female sex (Adjusted OR 2.34; 95%CI: 1.235–4.68; p = 0.001) and poor knowledge on the causes of overweight/obesity (Adjusted OR 2.15; 95%CI: 1.22–3.81; p = 0.008). In addition, DDS (Adjusted OR 0. 870; 95%CI: 0.76–0.99; p = 0.037) was negatively associated with overweight/obesity (Table 5).

**Table 2. Demographic characteristics of respondents (N = 313).**

| Variables | | Frequency | Percent |
|---|---|---|---|
| Age (years) | 25–49 | 222 | 70.9 |
| | 50–65 | 91 | 29.1 |
| Sex | Female | 196 | 62.6 |
| | Male | 117 | 37.4 |
| Marital Status | Married | 141 | 45.0 |
| | Never married | 71 | 22.7 |
| | Cohabiting | 34 | 10.9 |
| | Widow/widower | 30 | 9.6 |
| | Separated | 20 | 6.4 |
| | Divorced | 17 | 5.4 |
| Education level | Completed primary school | 168 | 53.7 |
| | Completed secondary school | 65 | 20.8 |
| | College/University | 36 | 11.5 |
| | Did not complete primary school | 19 | 6.1 |
| | Did not complete secondary school | 15 | 4.8 |
| | No formal schooling | 10 | 3.2 |
| Household size (>18 years) | 1–4 | 275 | 87.9 |
| | 5–8 | 35 | 11.2 |
| | >8 | 3 | 1.0 |
| Main occupation | Self employed | 209 | 66.8 |
| | House wife | 38 | 12.1 |
| | Non-Government employee | 33 | 10.5 |
| | Government employee | 16 | 5.1 |
| | Unemployed | 13 | 4.2 |
| | Retired officer | 2 | 0.6 |
| | Student | 2 | 0.6 |
| Source of income | Business | 171 | 54.6 |
| | Salary | 48 | 15.3 |
| | Aid | 27 | 8.6 |
| | Agriculture | 26 | 8.3 |
| | Labour selling | 23 | 7.3 |
| | Remittances | 15 | 4.8 |
| | Pension | 3 | 1.0 |
| Income earned per month (TSH) | ≤300,000 | 264 | 84.3 |
| | 301,000–600,000 | 30 | 9.6 |
| | 601,000–900,000 | 9 | 2.9 |
| | 901,000–1,200,000 | 6 | 1.9 |
| | >1,200,000 | 4 | 1.3 |
| Mean averages | Variables | N | Mean (SD) |
| | Age of respondent | 313 | 41.06 (±12.5) |
| | Average monthly income | 193 | 350,000 (±8,000) |
| | Household size (Adults ≥ 18 years) | 313 | 3(±1.7) |

## General knowledge about overweight and obesity

The results show that 75.7% (n = 237) of respondents reported to know the causes of overweight and obesity whereby 38.8% (n = 92) reported that the conditions are caused by high

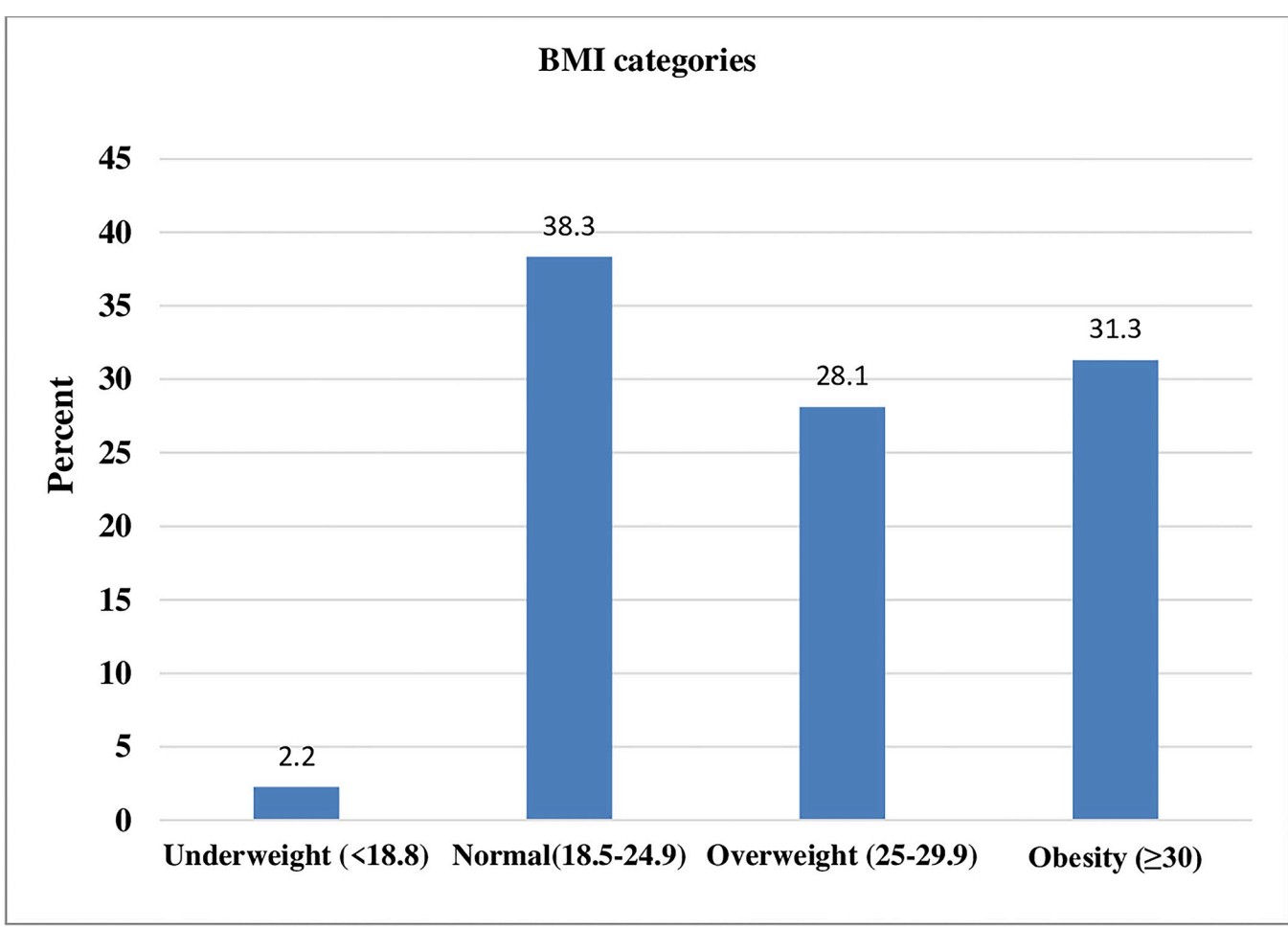

**Fig 1. Participants' nutrition status based on BMI (Kg/M$^2$).**

intake of energy foods, 26.6% (n = 92) intake of high fat foods while 13.5% (n = 32) mentioned both low physical activities and high fat intake to be the causes (Table 6).

## Knowledge about prevention of overweight and obesity

The results in Table 7 show that more than half (54%, n = 145) of the respondents are not sensitive with the kind of foods they consume. Among those who reported to be sensitive with what they eat, 64% reported to do so to avoid diseases while 9.7% (n = 14) reported to avoid

**Table 3. Nutrition statuses of participants based on sex.**

| Variables | Males (%) | Females (%) | Mean(SD) | P-Value |
|---|---|---|---|---|
| BMI(Kg/m$^2$) | | | | |
| Underweight (<18.5) | 5(4.3) | 2(1.0) | 27.3(±6.39) | 0.000 [**] |
| Normal (18–24.9) | 64(54.7) | 56(28.6) | | |
| Overweight (25–29.9) | 32(27.4) | 56(28.6) | | |
| Obese(≥30) | 16(13.7) | 82(41.8) | | |

**Note:** [**]Significant at p<0.05 and BMI bases on WHO criteria [1].

**Table 4. Food groups consumed within 24 hours.**

| Food group | Frequency | Percent |
|---|---|---|
| Cereals | 265 | 84.7 |
| White roots and tubers | 126 | 39.9 |
| Vitamin A rich vegetables and tubers | 208 | 66.5 |
| Dark green leafy vegetables | 154 | 49.2 |
| Other vegetables | 166 | 53 |
| Vitamin A rich fruits | 110 | 35.1 |
| Other fruits | 41 | 13.1 |
| Flesh meat | 19 | 6.1 |
| Organ meat | 8 | 2.6 |
| Eggs | 16 | 5.1 |
| Fish and sea foods | 76 | 24.3 |
| Legumes and nuts | 123 | 39.3 |
| Milk and milk products | 44 | 14.1 |
| Oils and fats | 313 | 100 |
| Sweets | 42 | 13.4 |
| Spices, condiments and beverages | 185 | 59.1 |
| Mean DDS | 313 | 4.7(SD ±1.7) |

**Note:** DDS = dietary diversity score, SD = Standard deviation

both diseases and overweight or obesity. Respondents knowledge on health foods show that 28.8% (n = 90) do not know foods that are good for their health while 37.4% (n = 117) mentioned health foods to be vegetables and fruits, and only 0.3% (n = 1) reported health foods to be those with low sugar. Furthermore, 42.8% (n = 134) do not know unhealthy foods at all, with 31.6% (n = 99) mentioning unhealthy food to be those with high fat content, 1.3% (n = 4) mentioned high energy foods and 2.9% mentioned unhealthy foods as those with high fat, salt and sugar.

## Practices against overweight and obesity

Results show that 56.2% (n = 176) of participants are concerned about their body weight. About 53.4% (n = 94) of those who reported to be concerned with their body weight said it is because they want to prevent themselves from diseases while 34.1% (n = 60) wanted to maintain their body weight so as to appear good. About 60% (n = 106) take care of their weight by doing physical exercises alone, followed by 17.65% (n = 31) who were managing their weight by doing both physical exercises and diet management. The practices of check their body weight shown that majority of the respondents (65.8%, n = 206) measure their body weight rarely, followed by those who do it monthly (13.7%, n = 43) (Table 8).

## Discussion

The current study was done to assess the prevalence, knowledge and practices on prevention and management of overweight/obesity among adults in Dodoma city, Tanzania to develop appropriate interventions to reduce the associated co-morbidities.

Findings of the study show the overall prevalence of overweight/obesity to be as higher as about 60% of which 28% were overweight and about 31% obese. The prevalence is higher among female compared to male participants. The reported high prevalence of overweight/obesity was positively associated with female sex which may be due to cultural aspects and

**Table 5. Risk factors for overweight/obesity based on BMI (Kg/M$^2$).**

| Variables | Crude OR | 95% CI | | Adjusted OR | 95% CI | |
|---|---|---|---|---|---|---|
| Sex | | | | | | |
| Male | 1 | | | 1 | | |
| Female | 3.505 | 2.17–5.67 | 0.000* | 2.34 | 1.235–4.68 | 0.001* |
| Age | | | | | | |
| 25–49 | 1 | | | | | |
| 50–65 | 1.778 | 1.06–2.99 | 0.03* | 0.669 | 0.38–1.78 | 0.163 |
| Number of adults ($\geq$ 18 years) in the household | | | | | | |
| 1–4 adults | 1 | | | | | |
| 5–8 adults | 0.96 | 0.62–7.76 | 0.77 | NA | | |
| >8 adults | 1.25 | 0.102–15.38 | 0.86 | NA | | |
| Knowledge on causes of overweight/obesity | | | | | | |
| Yes | 1 | | | 1 | | |
| No | 2.095 | 1.24–3.54 | 0.006* | 2.15 | 1.22–3.81 | 0.008* |
| Care on what to eat | | | | | | |
| Yes | 1 | | | | | |
| No | 1.020 | 0.648–1.605 | 0.932 | NA | | |
| Care body weight | | | | | | |
| Yes | 1 | | | | | |
| No | 0.955 | 0.61–1.51 | 0.843 | NA | | |
| Alcohol intake | | | | | | |
| No | 1 | | | | | |
| Yes | 1.277 | 0.811–2.011 | 0.291 | NA | | |
| Dietary diversity score | 0.513 | 0.291–0.906 | 0.021* | 0. 87 | 0.76–0.99 | 0.037* |

Note

*OR means Odd ratio; CI = confidence interval and NA = Not Applicable for multiple analysis, dietary diversity score was used as continuous variable. Also, the analysis included education levels and income but they were not associated with overweight/obesity.

preference of women appearance because among the participants who reported to be concerned with their body weight to ensure good appearance, majority was females. A similar study in Rwanda by Niyitegeka et al. [24] reported that in most areas of Africa, overweight/obesity is regarded as a sign of good nutrition, happiness and improved socio-economic conditions. The high prevalence of overweight/obesity was also reported in rural and semi-urban areas of Kilimanjaro to be a result of providing women with a leave of at least three months from household chores and other activities which may reduce their energy expenditure leading to increased prevalence of overweight and obesity [25]. Zubery et al. [26] who conducted a study in Arusha among adult workers reported a slightly higher prevalence of overweight/obesity of about 69% of which obesity (38%) was higher than overweight 31% as compared to the current study. The observed differences may be attributed to differences in socio economic status and lifestyles of the participants as the study involved formally employed workers (special group). However, this and the current study provide insight on the increase of the rate of obesity as compared to overweight which is exposing people to more health problems.

The reported high prevalence of overweight/obesity in the current study was also positively associated with poor knowledge on the risks and prevention of overweight/obesity even after adjusting for age of the respondents. Although majority of respondents reported to know the risks associated with overweight/obesity, only 14% mentioned both sedentary lifestyle and intake of energy dense foods as the risky foods. Furthermore, less than half of the participants

**Table 6. Respondents' general knowledge about overweight and obesity.**

| Variables | | Frequency | Percent |
|---|---|---|---|
| Do you know the causes of overweight and obesity? (N = 313) | Yes | 237 | 75.7 |
| | No | 76 | 24.3 |
| What causes overweight and obesity? (N = 237) | High intake of energy foods | 92 | 38.8 |
| | High fat intake | 63 | 26.6 |
| | Satisfaction and high fat foods | 40 | 16.9 |
| | Low physical exercise and high fat intake | 32 | 13.5 |
| | Low physical exercises | 6 | 2.5 |
| | Low intake of fruits and vegetables | 2 | 0.8 |
| | High intake of soft drinks like soda | 2 | 0.8 |
| What health related problems could occur when a person is overweight or obese? (N = 313) | Diabetes and Hypertension | 147 | 47.0 |
| | Hypertension | 94 | 30.0 |
| | Don't know | 60 | 19.2 |
| | Diabetes and Cancer | 5 | 1.6 |
| | Hypertension and Cancer | 4 | 1.3 |
| | Diabetes | 3 | 1.0 |
| Measures to prevent overweight and obesity (N = 313) | Engage/increase physical activities | 79 | 25.2 |
| | Reduce energy intake | 56 | 17.9 |
| | Low fat intake and increase activity | 52 | 16.6 |
| | Reduce intake of fat and sugar foods | 39 | 12.5 |
| | Don't know | 37 | 11.8 |
| | Eat vegetables and fruits more often | 28 | 8.9 |
| | Low fat intake | 22 | 7.0 |

mentioned the consequences of overweight/obesity as hypertension and diabetes while 19% did not know the impacts. It was noted further that nearly half of the participants were not concerned/worried about increase in body weight and for those who were concerned, more than half maintain their body weight through physical exercises alone and very few include both physical exercises and diet management. Although more than half of the participants are concerned with their increase in body weight, they rarely check their body weights. This implies that majority of the participants do not have adequate knowledge on how to control and/or manage their body weight. This is because, controlling weight by physical activity alone without considering other risk factors including diet, smoking, alcohol intake in the fight against overweight and obesity indicates low knowledge and poor practices in managing the condition. Some of the participants reported to be reducing the amount of foods they consume as a measure for addressing overweight/obesity without considering the types of food to be reduced and those which need to be increased whereby very few participants mentioned the reduction of fat foods and sugar intakes as a solution. This implies that participants had no enough knowledge on weight management to prevent overweight/obesity. Contrary to the current findings, a study done in Rwanda revealed that 22% of participants had higher knowledge and 62% had moderate knowledge on prevention of overweight and obesity [24]. The same study also reported that more than half of the participants have knowledge on different components for preventing overweight and obesity including the causes, whereby eating too much fat foods and insufficient physical activity were both mentioned as the causes of overweight and obesity which is again contrary to the current study whereby very few individuals have knowledge on prevention of overweight/obesity. The study further insists that to prevent overweight and obesity a combination of practices can be done including eating moderate food

**Table 7. Respondents' knowledge about prevention of overweight and obesity.**

| Variables | | Frequency | Percent |
|---|---|---|---|
| Do you care about type of food you eat? (N = 313) | Yes | 145 | 46.3 |
| | No | 168 | 53.7 |
| Why do you care about the type of food you eat? (N = 145) | To avoid diseases | 93 | 64.1 |
| | To avoid overweight/obesity | 35 | 24.1 |
| | To avoid diseases and being overweight/obesity | 14 | 9.7 |
| | Look good | 3 | 2.1 |
| Which foods are healthy generally? (N = 313) | Vegetables and/or fruits | 117 | 37.4 |
| | Don't know | 90 | 28.8 |
| | Milk and milk products | 27 | 8.6 |
| | Cereals | 19 | 6.1 |
| | Low energy foods (Unrefined foods) | 15 | 4.8 |
| | All foods are good for health | 15 | 4.8 |
| | Less fat foods | 10 | 3.2 |
| | Milk, fruits, vegetables, legumes | 7 | 2.2 |
| | Roots/tubers | 5 | 1.6 |
| | Low salt, fat and sugar foods | 3 | 1.0 |
| | High protein foods | 2 | 0.6 |
| | Legumes | 2 | 0.6 |
| | Less sugar foods | 1 | 0.3 |
| Which foods are generally unhealthy? (N = 313) | Don't know | 134 | 42.8 |
| | More fat foods | 99 | 31.6 |
| | Meat and meet products | 41 | 13.1 |
| | Much sugar foods | 23 | 7.3 |
| | High fat, salt and sugar foods | 9 | 2.9 |
| | High energy foods (Refined foods) | 4 | 1.3 |
| | Cereals | 3 | 1.0 |

with less fat, eating more fruits and vegetables, consumption of lower caloric drinks and avoiding alcoholic beverage consumption.

Additionally, overweight/obesity was negatively associated with DDS in the current study meaning that decrease in DDS increases the rate of overweight/obesity although the mean DDS was found to be moderate (that is almost average of 5 food groups were consumed in a day). The study also found that the mostly consumed food groups were oil and fats which were reported by all participants followed by cereals, Vitamin A rich vegetables and tubers as well as spices, condiments and beverages of which salt, sugar and carbonated soft drinks were leading. Most of the highly consumed foods are rich sources of energy which may have contributed to their weight gain and finally overweight/obesity which also implies that high DDS does not always guarantee quality dietary intake. A study done in Monduli supports our findings that 54.3% of the participants consumed an adequate dietary diversity (DDS ≥4) but it was inconsistently positively associated with overweight/obesity among males and females which may be attributed by the most consumed foods which were dominated by cereals or legumes staples, with moderate intake of animal sourced foods, and very little fruits [27]. Another study among Whites, Hispanic, Blacks, and Chinese supports our findings that higher DDS does not necessarily guarantee healthy eating [28]. This is because the individuals may be consistently consuming diverse food groups which are unhealthy including red meat, processed meat, potatoes, refined grains and baked goods, sugar-sweetened beverages, fried foods eaten away

**Table 8. Respondents' practices against overweight and obesity.**

| Variables | | Frequency | Percent |
|---|---|---|---|
| Do you care about your body weight? (N = 313) | Yes | 176 | 56.2 |
| | No | 137 | 43.8 |
| Why do you care about your body weight? (N = 176) | To avoid diseases | 94 | 53.4 |
| | To look good | 60 | 34.1 |
| | Good looking and avoid diseases | 10 | 5.7 |
| | Increase work performance (Activeness) | 5 | 2.8 |
| | Avoid overweight/obesity | 5 | 2.8 |
| | Advised by doctor | 2 | 1.1 |
| What measures do you take to care for your body weight (N = 176) | Doing exercises | 106 | 60.2 |
| | Reduce intake of high fat and sugar foods | 9 | 5.1 |
| | Reduce the amount of food | 15 | 8.5 |
| | Reduce intake of fatty foods | 15 | 8.5 |
| | Diet management and physical activities | 31 | 17.6 |
| How often do you measure your body weight? (N = 313) | Rarely | 206 | 65.8 |
| | Monthly | 43 | 13.7 |
| | Weekly | 36 | 11.5 |
| | Never | 26 | 8.3 |
| | Every time I go to hospital | 2 | 0.6 |

from home, as well as sweets and ice cream. Therefore, there is a need to promote healthy eating patterns emphasizing on adequate consumption of plant foods, protein sources, low-fat dairy products, vegetable oils, and nuts while limiting the intake of sweets, sugar-sweetened beverages, and red meats as preventive measures for overweight/obesity [29].

## Conclusion and recommendations

The study findings indicate a high prevalence of overweight and obesity, which may increase the risk of developing diet related non-communicable diseases in the study area. The observed prevalence was significantly associated with low DDS, knowledge on the causes of overweight/obesity and female sex. Low knowledge on different aspects of overweight/obesity was also reported in this study including the risk factors, consequences, controlling and preventive measures, which may affect self-care practices. Therefore, there is a need to develop a training program that can help improve adults' knowledge on overweight and obesity for enhancing self-care to reduce the observed prevalence and NCDs risk as a whole. Although the intervention will target all population groups, special attention should be given to female adults in the study area since they are more likely to encounter overweight and obesity as compared to their male counterparts.

## Supporting information

**S1 Data.**
(XLSX)

## Author Contributions

**Conceptualization:** Safiness Simon Msollo, Gosbert Lukenku Shausi, Akwilina Wendelin Mwanri.

**Data curation:** Safiness Simon Msollo, Gosbert Lukenku Shausi.

**Formal analysis:** Safiness Simon Msollo, Gosbert Lukenku Shausi, Akwilina Wendelin Mwanri.

**Methodology:** Safiness Simon Msollo, Gosbert Lukenku Shausi, Akwilina Wendelin Mwanri.

**Project administration:** Safiness Simon Msollo.

**Software:** Gosbert Lukenku Shausi.

**Supervision:** Akwilina Wendelin Mwanri.

**Writing – original draft:** Safiness Simon Msollo.

**Writing – review & editing:** Safiness Simon Msollo, Gosbert Lukenku Shausi, Akwilina Wendelin Mwanri.

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
