## [Decision Letter · Decision Letter 0]

20 Feb 2023

PONE-D-22-34731Prevalence, Knowledge and Practices on Prevention and Management of Overweight and Obesity among Adults in Dodoma City, TanzaniaPLOS ONE

Dear Dr. Msollo,

Thank you for submitting your manuscript to PLOS ONE. After careful consideration, we feel that it has merit but does not fully meet PLOS ONE’s publication criteria as it currently stands. Therefore, we invite you to submit a revised version of the manuscript that addresses the points raised during the review process.

We look forward to receiving your revised manuscript.

Kind regards,

Andrea Maugeri

Academic Editor

PLOS ONE

Journal Requirements:

Reviewers' comments:

Reviewer's Responses to Questions

**Comments to the Author**

1. Is the manuscript technically sound, and do the data support the conclusions?

Reviewer #1: Yes

Reviewer #2: Yes

2. Has the statistical analysis been performed appropriately and rigorously? 

Reviewer #1: Yes

Reviewer #2: Yes

3. Have the authors made all data underlying the findings in their manuscript fully available?

Reviewer #1: Yes

Reviewer #2: Yes

4. Is the manuscript presented in an intelligible fashion and written in standard English?

Reviewer #1: Yes

Reviewer #2: Yes

5. Review Comments to the Author

Reviewer #1: The authors aimed to establish the knowledge gaps on the prevention and management for overweight and obesity among adults in Dodoma City, Tanzaniaas, an evidence for designing an intervention.

The introduction is clear, however, I suggest to briefly describe the potential role of molecular mechanisms involved in the relationship between social determinants, lifestyles and health. Consider, for instance, the following study:

doi: 10.1155/2021/9910878.

Moreover, could the authors better define the inclusion and exclusion criteria? Were there also pregnant women? Pregnant women could benefit from ad hoc interventions for , confirming the need to better understand the determinants of health involved in their nutritional choices and obesity risk (10.7416/ai.2019.2280; https://doi.org/10.1186/1756-0500-6-278)

The authors reported that "the assessed foods were grouped into 16 food groups to obtain their

dietary diversity score (DDS)". Could the authors better explain the methods used for DDS?

In the discussion I suggest to include a section of limitation, including the use of 24h recall.

Reviewer #2: The authors report a study on the Prevalence, Knowledge, and Practices on Prevention and Management of Overweight and Obesity among Adults in Dodoma City, Tanzania. The main objective of this analysis was to establish the knowledge gaps on the prevention and management of overweight and obesity among adults as evidence for designing an intervention. I have several concerns that should be addressed

General

The authors need to improve the English language within this manuscript, especially in the discussion section. I recommend that they either get their manuscript reviewed by someone who is fluent in English or, if they would like professional help, they can use any reputable English language editing service. I have identified several problems with grammar/spelling/punctuation/language.

Materials and methods

Description of the study area, population and study design

In this section, the authors should update their information by using results from the 2022 population and housing census, as this is currently available.

Sample size determination

The authors started using the term DR-NCDs without first explaining the meaning of it

Results

The authors mention that smoking was assessed using a questionnaire with closed and open-ended questions adapted from the STEPS instrument for NCDs. However, throughout the result section, I could not see smoking mentioned. I suggest not mentioning smoking in the Demographic characteristics section if its data is not used during the analysis.

Demographic characteristics of the participants

Table 1 is referred to in this section. However, I think there is confusion as the reference made does not tally with what is presented in table 1.

Discussion

Can the author state clearly when are women in the Kilimanjaro area provided with a leave of at least three months from household chores and other activities? In the reference they used; it is indicated that this happens soon after a woman delivers. Also, are the authors suggesting that this could be an explanation for the high prevalence of overweight/obesity positively associated with the female sex in the current study area as well?

In the third paragraph, there seems to be a repetition of the results which is unnecessary. The authors should focus more on explaining (discussing) the results

Conclusion and Recommendations

The authors are recommending a training program that can help improve adults’ knowledge of overweight and obesity for enhancing self-care to reduce the observed prevalence and NCDs risk. However, I suggest the authors state exactly how that training program should be implemented. For example, they could state that it will be a public educational campaigning

6. PLOS authors have the option to publish the peer review history of their article (what does this mean?). If published, this will include your full peer review and any attached files.

Reviewer #1: No

Reviewer #2: No

---

## [Author Response · Author response to Decision Letter 0]

19 Apr 2023

Reviewers Comments Responses Page No 

Editor Thank you for submitting your manuscript to PLOS ONE. After careful consideration, we feel that it has merit but does not fully meet PLOS ONE’s publication criteria as it currently stands. Therefore, we invite you to submit a revised version of the manuscript that addresses the points raised during the review process. Thank you for seen the importance of this manuscript 

The comments are addressed and highlighted yellow in the main text NA

 A rebuttal letter that responds to each point raised by the academic editor and reviewer(s). You should upload this letter as a separate file labeled 'Response to Reviewers' This has been included as attached NA

 A marked-up copy of your manuscript that highlights changes made to the original version. You should upload this as a separate file labeled 'Revised Manuscript with Track Changes'. This has been addressed as suggested NA

 An unmarked version of your revised paper without tracked changes. You should upload this as a separate file labeled 'Manuscript'. Attached as suggested NA

 Please ensure that your manuscript meets PLOS ONE's style requirements, including those for file naming This has been considered as suggested NA

 We note that the grant information you provided in the ‘Funding Information’ and ‘Financial Disclosure’ sections do not match. When you resubmit, please ensure that you provide the correct grant numbers for the awards you received for your study in the ‘Funding Information’ section. The award number has been included Pg 20

 In your Data Availability statement, you have not specified where the minimal data set underlying the results described in your manuscript can be found. PLOS defines a study's minimal data set as the underlying data used to reach the conclusions drawn in the manuscript and any additional data required to replicate the reported study findings in their entirety. All PLOS journals require that the minimal data set be made fully available. Upon re-submitting your revised manuscript, please upload your study’s minimal underlying data set as either Supporting Information files or to a stable, public repository and include the relevant URLs, DOIs, or accession numbers within your revised cover letter. The statement regarding data availability is included. Minimal data set will be made available upon request to the corresponding author. However, data sharing at this stage is difficult as this study is part of ongoing study Pg. 20

 Important: If there are ethical or legal restrictions to sharing your data publicly, please explain these restrictions in detail. Note that it is not acceptable for the authors to be the sole named individuals responsible for ensuring data access. We will update your Data Availability statement to reflect the information you provide in your cover letter. There is no any restriction on data sharing however, as this study is part of a large study, there are other unprocessed data which cannot be shared because the study is still going on Pg. 20

 PLOS requires an ORCID iD for the corresponding author in Editorial Manager on papers submitted after December 6th, 2016. Please ensure that you have an ORCID iD and that it is validated in Editorial Manager. To do this, go to ‘Update my Information’ (in the upper left-hand corner of the main menu), and click on the Fetch/Validate link next to the ORCID field https://orcid.org/0000-0003-0646-2313 Pg. 1

 Your ethics statement should only appear in the Methods section of your manuscript. If your ethics statement is written in any section besides the Methods, please move it to the Methods section and delete it from any other section. Please ensure that your ethics statement is included in your manuscript, as the ethics statement entered into the online submission form will not be published alongside your manuscript. This statement has been shifted to methods section as suggested Pg 9

 Please review your reference list to ensure that it is complete and correct. If you have cited papers that have been retracted, please include the rationale for doing so in the manuscript text, or remove these references and replace them with relevant current references. Any changes to the reference list should be mentioned in the rebuttal letter that accompanies your revised manuscript. If you need to cite a retracted article, indicate the article’s retracted status in the References list and also include a citation and full reference for the retraction notice. References have been well reviewed as suggested Pg. 21-25

Reviewer 1 and 2 Is the manuscript technically sound, and do the data support the conclusions? Yes 

Has the statistical analysis been performed appropriately and rigorously? Yes Thank you for seeing the soundness of the study NA

 Have the authors made all data underlying the findings in their manuscript fully available? Yes Thank you for seeing the inclusion of the required data in the manuscript Pg. 10-15

 Is the manuscript presented in an intelligible fashion and written in standard English? Yes Thank you for appreciating this NA

Reviewer #1 The authors aimed to establish the knowledge gaps on the prevention and management for overweight and obesity among adults in Dodoma City, Tanzania as, an evidence for designing an intervention. The introduction is clear, however, I suggest to briefly describe the potential role of molecular mechanisms involved in the relationship between social determinants, lifestyles and health. Consider, for instance, the following study:

doi: 10.1155/2021/9910878. A brief description of the potential role of molecular mechanisms has been included Pg. 2&3

 Moreover, could the authors better define the inclusion and exclusion criteria? Were there also pregnant women? Pregnant women could benefit from ad hoc interventions for , confirming the need to better understand the determinants of health involved in their nutritional choices and obesity risk (10.7416/ai.2019.2280; https://doi.org/10.1186/1756-0500-6-278) As the study used BMI as a measure for overweight/obesity, pregnant women could not fit in the study however, as the study aims to establish knowledge gaps, the intervention will include all adults regardless of their physiological status Pg. 4

 The authors reported that "the assessed foods were grouped into 16 food groups to obtain their

dietary diversity score (DDS)". Could the authors better explain the methods used for DDS? This has been well explained in the dietary assessment section (Ref. Guideline for measuring individual and household dietary diversity https://www.fao.org/3/i1983e/i1983e00.pdf Pg. 7

 In the discussion I suggest to include a section of limitation, including the use of 24h recall. The limitation statement has been included as a single 24 hours recall was used. The dietary diversity score as a proxy for nutrient adequacy is mainly validated for women and children. Pg. 18

Reviewer #2 The authors report a study on the Prevalence, Knowledge, and Practices on Prevention and Management of Overweight and Obesity among Adults in Dodoma City, Tanzania. The main objective of this analysis was to establish the knowledge gaps on the prevention and management of overweight and obesity among adults as evidence for designing an intervention. I have several concerns that should be addressed : General: The authors need to improve the English language within this manuscript, especially in the discussion section. I recommend that they either get their manuscript reviewed by someone who is fluent in English or, if they would like professional help, they can use any reputable English language editing service. I have identified several problems with grammar/spelling/ punctuation/language. 

The manuscript has been reviewed by an English fluent person as suggested 

Pg 1-25

 Materials and methods

Description of the study area, population and study design

In this section, the authors should update their information by using results from the 2022 population and housing census, as this is currently available. Sample size determination The data have updated for region but data for district level is yet to be released Pg. 4

 The authors started using the term DR-NCDs without first explaining the meaning of it The word has been changed to NCDs which is commonly used 

 Results

The authors mention that smoking was assessed using a questionnaire with closed and open-ended questions adapted from the STEPS instrument for NCDs. However, throughout the result section, I could not see smoking mentioned. I suggest not mentioning smoking in the Demographic characteristics section if its data is not used during the analysis. This was overlooked as it was included in the main study but not in this specific study hence, it has been deleted as suggested Pg. 6

 Demographic characteristics of the participants

Table 1 is referred to in this section. However, I think there is confusion as the reference made does not tally with what is presented in table 1. The correct citation of the table has been done Pg. 10

 Discussion

Can the author state clearly when are women in the Kilimanjaro area provided with a leave of at least three months from household chores and other activities? In the reference they used; it is indicated that this happens soon after a woman delivers. Also, are the authors suggesting that this could be an explanation for the high prevalence of overweight/obesity positively associated with the female sex in the current study area as well? This has been clarified and it is used as one of the factors that could contribute to the high prevalence of overweight/obesity among females which need further explorations to get the practices in the study area as well. Pg. 16

 In the third paragraph, there seems to be a repetition of the results which is unnecessary. The authors should focus more on explaining (discussing) the results The repeated results have been deleted Pg. 16-17

 Conclusion and Recommendations 

The authors are recommending a training program that can help improve adults’ knowledge of overweight and obesity for enhancing self-care to reduce the observed prevalence and NCDs risk. However, I suggest the authors state exactly how that training program should be implemented. For example, they could state that it will be a public educational campaigning This has been clarified as suggested in the conclusion and recommendations section Pg 18-19

Reviewer 3 Recommendation -Reject

 The recommendation has been clarified Pg 18-19

 Is the manuscript technically sound, and do the data support the conclusions?-No We think that the data are enough to give the conclusion based on the aim of the study Pg 10-15, 18-19

 Has the statistical analysis been performed appropriately and rigorously? (Answer

Options-No According to the data needed to make conclusion, the analysis done is appropriate Pg. 9

 Does the manuscript adhere to the PLOS Data Policy? Additional details can be found at ttp://journals.plos.org/plosone/s/materials-and-software-sharing. (Answer options:-Yes Thank you for agreeing with us NA

 Is the manuscript presented in an intelligible fashion and written in standard English?-Yes Thank you for agreeing with NA

 It is wrong using a prevalence of hypertension in determining the sample size for overweight/obesity study, considering they are two different conditions. Need to use the right formula for community surveys 

Recommendation: Re-do the sample size estimation, and consider recruiting

additional participants if the new sample size exceed the prior one. Need to use regional or national prevalence of overweight/obesity in sample size estimation because it is available in the previous national surveys. The sample size was calculated based on hypertension prevalence as the study is part of the large study dealing with improving knowledge on NCDs. Pg. 5

 Collection of 24 hours dietary recall information is recommended to be carried in at least two non-consecutive days to allow actual estimation of usual dietary intake. This study is highly biased on dietary estimates because of conducting only one time 24 hrs dietary recall interview. Recommendation: Need to consider conducting another round of 24 hours dietary recall interviews in the same study participants This has been indicated as a limitation of the study hence, no need for another round of data collection 

 Pg 18

---

## [Decision Letter · Decision Letter 1]

14 Jun 2023

PONE-D-22-34731R1Prevalence, Knowledge and Practices on Prevention and Management of Overweight and Obesity among Adults in Dodoma City, TanzaniaPLOS ONE

Dear Dr. Safiness Msollo,

Thank you for submitting your manuscript to PLOS ONE. After careful consideration, we feel that it has merit but does not fully meet PLOS ONE’s publication criteria as it currently stands. Therefore, we invite you to submit a revised version of the manuscript that addresses the points raised during the review process.

When responding to reviewers' comments, please make sure to thoroughly  respond to previous reviewer comments, including justifications  for choice of study area//conducting same study (as previous one) in the same area, clear methods/ and presentation of respective results, and in depth discussion of your results (with possible involved mechanisms behind associated risk factors). 

We look forward to receiving your revised manuscript.

Kind regards,

Elingarami Sauli, PhD

Academic Editor

PLOS ONE

Journal Requirements:

Reviewers' comments:

Reviewer's Responses to Questions

**Comments to the Author**

1. If the authors have adequately addressed your comments raised in a previous round of review and you feel that this manuscript is now acceptable for publication, you may indicate that here to bypass the “Comments to the Author” section, enter your conflict of interest statement in the “Confidential to Editor” section, and submit your "Accept" recommendation.

Reviewer #2: All comments have been addressed

Reviewer #3: (No Response)

Reviewer #4: (No Response)

2. Is the manuscript technically sound, and do the data support the conclusions?

Reviewer #2: (No Response)

Reviewer #3: Yes

Reviewer #4: Partly

3. Has the statistical analysis been performed appropriately and rigorously? 

Reviewer #2: (No Response)

Reviewer #3: Yes

Reviewer #4: I Don't Know

4. Have the authors made all data underlying the findings in their manuscript fully available?

Reviewer #2: (No Response)

Reviewer #3: No

Reviewer #4: Yes

5. Is the manuscript presented in an intelligible fashion and written in standard English?

Reviewer #2: (No Response)

Reviewer #3: Yes

Reviewer #4: Yes

6. Review Comments to the Author

Reviewer #2: (No Response)

Reviewer #3: Manucript title: Prevalence, Knowledge and Practices on Prevention and Management of Overweight and Obesity among Adults in Dodoma City, Tanzania

GENERAL COMMENTS

The other published work about prevalence of overweight and obesity among adults in dodoma was conducted by Mariam Munyogwa (doi: 10.1155/2018/6123156). The author should justify why this study is repeated.

When you write overweight/obesity, it is confusing which one do you mean. Overweight is not the same as obesity. Either use obesity and overweight or find the best way to describe the two groups.

OTHER COMMENTS

Abstract

― Use the word “24-hours diet recall” and not “24 hours recall”

― I think just say the prevalence of overweight and obesity was 59.7%, to avoid confusion.

― Put bracket at the end “Obesity was three times higher among females than males (41.8% vs 13.7%)”.

― Only two decimal places is enough (Adjusted OR 2.34; 95%CI: 1.23-4.68; p=0.001). correct

― Be consistent if you use percent and number into the bracket, follow that order. Example 28% (n=88). Some where you present as (27%; n=63), and some didn’t show number(n).

― In conclusion, by saying high prevalence was observed can be inappropriate term. Just use word such as one out of two were overweight and obese.

― In abstract prevalence is 59% while in results is 60%.

Introduction

By starting with the definition of overweight and obesity is not giving much information about the problem. Better show statistics in terms of global trends, regional and in the country. The information shown in the first and second paragraph are known and is just repeating what has been published before. Like the cut off points could be shown in the methods.

The author previously mention the prevalence of abdominal obesity in Dodoma region. Than, mentioned that there is paucity of data. More clarification is needed here, and show the gaps and weakness of the previous studies.

Literature review is weak, more evidences are needed to justify the problem being investigated, importance of conducting this study and gaps.

Materials and Methods

Combining description of the study, population and study design is not good approach. Consider separating those sub-heading.

p=proportion of people with NCD? What does that mean, which condition was considered. Just say if hypertension, obesity etc.

Who coded the food items into groups? And how many were involved in the coding?

In ethical consideration, some information about consent forms were already explained in the data collection methods.

Table 1: I think you don’t need this table here, it is known you can just cite FAO

In data analysis what conditions did you consider in putting the variable in multiple logistic regression?

Results

Again, be consistent in format of percentage and no. choose one style

About 71% were adult of reproductive age (25-49). This may mislead. Just say 71% were aged between 25 to 49 years.

The mean age is for whole sample of just 25-49 years?

Sub-heading of Nutrition status just say “ Prevalence of overweight and obesity”

Table 3: Title should be prevalence of overweight and obesity

I advice Table 6 and 7 should be combined, because all talking about knowledge.

Discussion

The whole discussion section needs revision, especially explain points in paragraphs. Each paragraph must have a specific issue to talk about. Compare contrast, explain the limitations in your methodology and implication of the results to the community.

Reviewer #4: Overall comments:

The English is still not at an adequate level and should be revised again. The manuscript is too long and can be shortened easily. Reviewer comments should be addressed in the manuscript, not just in the answers to the comments, several requests have not been addressed. The sample size estimation can easily be fixed to address the previous reviewer’s comment (see below for suggestion). Line numbers would have allowed me to comment on specific words/sentences and suggest improvements.

Specific comments:

Abstract:

The abstract does not reflect the paper and should be revised to include more specific info such as the fact the study was done in Tanzania, that the aim was to inform interventions and regression was done to see which factors were associated with overweight/obesity.

Introduction:

Too long. Paragraphs 1 and 2 can be merged and summarized. You need to answer the following: Why is it important that Dodoma data is known? Why would your results be different from studies done in Dar? You need to justify that. For example with the fact that Dodoma is an example of a fast growing economy, fast growing population and looked at as a model city because of its administrative importance for the country. An intervention here may be easier to pilot because of the physical presence of the MoH there.

I am missing the cultural aspect introduced in the paper that is mentioned in the discussion on the fact that in Tanzania, as in many African countries, bigger body sizes are aspired to, especially for women and babies, and are mistaken for a sign of health and having achieved a good life. For men, being overweight is synonymous with a good marriage and comfortable life. This makes it hard to change perceptions.

Materials and Methods:

Overall: The methods section has to be revised thoroughly. Comments from a previous review were not all addressed. Smoking is still in there even though it is not used in this study. Selection criteria are not clear. Analysis is not clear. More data collected than used in the regression.

I suggest you reference well-established methods and guidelines instead of explaining how to measure height in detail. To improve your methods section, please look at your reference 27 (Khamis et al. ) and at the following paper: Gibson Benard Kagaruki, Michael J Mahande, Godfather D Kimaro, Esther S Ngadaya, Mary T Mayige, Judith Msovela, Katharina Kreppel, Sayoki G Mfinanga & Bassirou Bonfoh (2022) Knowledge and attitudes towards type 2 diabetes and prevention strategies among regular street food consumers: A cross sectional study in Dar es Salaam, Tanzania, International Journal of Health Promotion and Education, DOI: 10.1080/14635240.2022.2104742

Specifically:

The study area part should include important information such as common livelihood of the urban population in Dodoma, e.g. business. Due to the relatively recent relocation of many governmental and non-governmental organisations to the capital from Dar es salaam, population and economic growth increased in the last years. Mention this.

Sample size: see Kagaruki et al (cited below) regarding your sample size calculation, as the previous reviewer was right and hypertension prevalence is not a good replacement. Recalculate your sample size with Kagaruki’s method (your sample size should be enough) and explanation.

Please explain the abbreviation DR-NCD.

If the response rate expected is 95% you don’t need to mention that you expect a 5% non-response rate!

Training of enumerators:

Info is missing: how many? Can they speak Swahili? How big is each team? One team or more? Explain Morogoro context like: “Administration of questionnaires and dietary recall in the official language Kiswahili, was piloted using 20 volunteers in the city of Morogoro situated in another region of Tanzania, but with similar characteristics.”

State your selection criteria! Adult 18+ years, male or female, not pregnant, not suffering from a known disease, etc.

You cannot say : “requested to sign” instead you obtained informed written consent and all participants were assured they can leave the study at any time without consequences.

ID given to link questionnaire and recall data with anthropometric measurements.

Were interviews audio recorded? Transcribed? Translated? Who did quality control?

DDS: table 1 seems superfluous. Simply list the 16 food groups and reference the FAO.

The way you write it: “respondents’ views on the causes, consequences, and risk factors…” it sounds like a KAP study looking at attitudes as well as knowledge and practices. Views are attitudes, not knowledge. Adjust please. Did you do a KAP study or did you ask for knowledge and practices only?

Data analysis:

Look at other papers to improve your data analysis section. Which variables were independent? Binomial distribution etc. Which did you use in the analysis. Why did you not use all? That needs to be justified.

This is important, because you did not use all the factors you assessed! How did you include marital status? You should only have living with a partner or not living with a partner, because from a health perspective you can argue that there’s no difference between married or co-habiting.

You adjusted for age, so you have to say here what variables you used, how you combined them etc.

Results:

Demographics: education is seen as important when considering knowledge, so the text should reflect how many have primary, secondary or more.

Monthly income in USD please

Labour selling – revise term (day labourer)

Why is the average income only for 193 participants? What do the other 16 live off?

Table 3 only shows underweight to be significantly higher in males – there’s no significant difference for the others. Explain?

Revise this sentence and split it up, it makes no sense:

“The results show that the mean DDS for the study population was 4.7 food groups where the

most consumed food groups in 24 hours were oil and fats which were consumed by all cereals

265(84.8%), Vitamin A rich vegetables and tubers 208 (66.5%) followed by spices, condiments

and beverages (185(59.1%) of which salt and soft drinks were highly consumed in this group”

It would be interesting to know if more females or males care about their health and what the different numbers are regarding exercise (more males?).

Discussion:

You also cannot simply mention location names without saying in which country they are. The Monduli study was on a specific population (pastoralist) and in a rural area and cannot be compared to your study without mentioning that!

The section on limitations should include the lack of questions about body image (fat is healthy), food availability and beliefs of what is a good diet.

7. PLOS authors have the option to publish the peer review history of their article (what does this mean?). If published, this will include your full peer review and any attached files.

Reviewer #2: No

Reviewer #3: No

Reviewer #4: No

---

## [Author Response · Author response to Decision Letter 1]

26 Oct 2023

Overall comments:

Comment: The English is still not at an adequate level and should be revised again. The manuscript is too long and can be shortened easily. 

Response: The English has been improved and poof read by an English native professor as well as shortened as suggested 

Comment: Reviewer comments should be addressed in the manuscript, not just in the answers to the comments; several requests have not been addressed

Response: responses have been incorporated in the manuscript as suggested 

Comment: The sample size estimation can easily be fixed to address the previous reviewer’s comment (see below for suggestion) 

Response: This has been well explained as the study is part of a large study on NCDS 

Comment: Line numbers would have allowed me to comment on specific words/sentences and suggest improvements. 

Response: Line numbers have been used

Specific comments:

Abstract:

The abstract does not reflect the paper and should be revised to include more specific info such as the fact the study was done in Tanzania, that the aim was to inform interventions and regression was done to see which factors were associated with overweight/obesity. 

Response: This has been addressed accordingly (Line 27-36)

Introduction

Comment: Too long. Paragraphs 1 and 2 can be merged and summarized . 

Response: The introduction has been summarized to be short as suggested (Line 42-51)

Comment: You need to answer the following: Why is it important that Dodoma data is known? 

Response: This has been addressed in the description of the study sub-section of the methodology (Line 89-95)

Comment: Why would your results be different from studies done in Dar? You need to justify that. For example with the fact that Dodoma is an example of a fast growing economy, fast growing population and looked at as a model city because of its administrative importance for the country. An intervention here may be easier to pilot because of the physical presence of the MoH there.

Response: Has been addressed as suggested (Line 92-95)

 Materials and Methods:

Comment: Overall: The methods section has to be revised thoroughly. Comments from a previous review were not all addressed. Smoking is still in there even though it is not used in this study. 

Response: Smoking is no longer found in the manuscript as it was removed in the second version of the manuscript 

Comment: Selection criteria are not clear. Analysis is not clear. More data collected than used in the regression.

Response: The analysis used many variables but they were not significant hence were removed during analysis as stated in the “note raw’ (Line 199-202 and 256)

Comment: I suggest you reference well-established methods and guidelines instead of explaining how to measure height in detail. To improve your methods section, please look at your reference 27 (Khamis et al. ) and at the following paper: Gibson Benard Kagaruki, Michael J Mahande, Godfather D Kimaro, Esther S Ngadaya, Mary T Mayige, Judith Msovela, Katharina Kreppel, Sayoki G Mfinanga & Bassirou Bonfoh (2022) Knowledge and attitudes towards type 2 diabetes and prevention strategies among regular street food consumers: A cross sectional study in Dar es Salaam, Tanzania, International Journal of Health Promotion and Education, DOI: 10.1080/14635240.2022.2104742

Response: The section has been improved as suggested (Line 153-160)

Comment : The study area part should include important information such as common livelihood of the urban population in Dodoma, e.g. business. Due to the relatively recent relocation of many governmental and non-governmental organisations to the capital from Dar es salaam, population and economic growth increased in the last years. Mention this.

Response: Addressed as suggested (Line 93-95)

Comment: Sample size: see Kagaruki et al (cited below) regarding your sample size calculation, as the previous reviewer was right and hypertension prevalence is not a good replacement. Recalculate your sample size with Kagaruki’s method (your sample size should be enough) and explanation.

Response: This was addressed in the first round as the study is part of the large study about NCDs (Line 124-128)

Comment: If the response rate expected is 95% you don’t need to mention that you expect a 5% non-response rate!

Response: Has been addressed as suggested (Line 125)

Training of enumerators:

Comment: Info is missing: how many? Can they speak Swahili? How big is each team? One team or more? Explain Morogoro context like: “Administration of questionnaires and dietary recall in the official language Kiswahili, was piloted using 20 volunteers in the city of Morogoro situated in another region of Tanzania, but with similar characteristics.”

Response: This has been addressed as suggested (Line 131-138)

Comment: State your selection criteria! Adult 18+ years, male or female, not pregnant, not suffering from a known disease, etc.

Response: The selection criteria was addressed as suggested (Line 96-100)

Comment: You cannot say: “requested to sign” instead you obtained informed written consent and all participants were assured they can leave the study at any time without consequences. 

Response: Corrected in ethical consideration as suggested and the ID given to link questionnaire and recall data with anthropometric measurements. (Line 140-143)

Comment: DDS: table 1 seems superfluous. Simply list the 16 food groups and reference the FAO.

Response: This has been addressed as suggested (Line 171-175)

Comment: The way you write it: “respondents’ views on the causes, consequences, and risk factors…” it sounds like a KAP study looking at attitudes as well as knowledge and practices. Views are attitudes, not knowledge. Adjust please. Did you do a KAP study or did you ask for knowledge and practices only?

Response: We asked for knowledge and practices hence the term views was changed to knowledge (Line 182)

Data analysis: 

Comment: Look at other papers to improve your data analysis section. Which variables were independent? Binomial distribution etc. Which did you use in the analysis. Why did you not use all? That needs to be justified.

Response: This has been stated clearly in the note bellow the table of results as well as in the data analysis section (Line 193-205)

Comment: This is important, because you did not use all the factors you assessed! How did you include marital status? You should only have living with a partner or not living with a partner, because from a health perspective you can argue that there’s no difference between married or co-habiting.

Response: This has been addressed as suggested however, no significant difference was observed (Line 256)

Comment: You adjusted for age, so you have to say here what variables you used, how you combined them etc.

Response: Addressed as suggested (Line 256)

Results:

Comment” Demographics: education is seen as important when considering knowledge, so the text should reflect how many have primary, secondary or more.

Responses: This has been addressed accordingly (Line 224)

Comment: Monthly income in USD please

Response: Changes into USD as suggested (Line 224)

Comment: Labour selling – revise term (day labourer)

Response: This has been revised as suggested (Line 224)

Comment: Why is the average income only for 193 participants? What do the other 16 live off?

Response: Some participants did not agreed to mention their average income (Line 224)

Comment: Table 3 only shows underweight to be significantly higher in males – there’s no significant difference for the others. Explain?

Response: This has been corrected for clarity (Line 233)

Comment: Revise this sentence and split it up, it makes no sense: 

“The results show that the mean DDS for the study population was 4.7 food groups where the

most consumed food groups in 24 hours were oil and fats which were consumed by all cereals

265(84.8%), Vitamin A rich vegetables and tubers 208 (66.5%) followed by spices, condiments

and beverages (185(59.1%) of which salt and soft drinks were highly consumed in this group”

Response: The sentence has been splitted as suggested (Line 240-244)

Discussion:

Comment: You also cannot simply mention location names without saying in which country they are. The Monduli study was on a specific population (pastoralist) and in a rural area and cannot be compared to your study without mentioning that! 

Response: This has been addressed accordingly (Line 299-383)

Comment: The section on limitations should include the lack of questions about body image (fat is healthy), food availability and beliefs of what is a good diet.

Response: This has been included (Line 384-389)

---

## [Decision Letter · Decision Letter 2]

10 Jan 2024

Prevalence, Knowledge and Practices on Prevention and Management of Overweight and Obesity among Adults in Dodoma City, Tanzania

PONE-D-22-34731R2

Dear Dr. Safiness Msollo,

We’re pleased to inform you that your manuscript has been judged scientifically suitable for publication and will be formally accepted for publication once it meets all outstanding technical requirements.

Kind regards,

Elingarami Sauli, PhD

Academic Editor

PLOS ONE

Additional Editor Comments (optional):

Reviewers' comments:

Reviewer's Responses to Questions

**Comments to the Author**

1. If the authors have adequately addressed your comments raised in a previous round of review and you feel that this manuscript is now acceptable for publication, you may indicate that here to bypass the “Comments to the Author” section, enter your conflict of interest statement in the “Confidential to Editor” section, and submit your "Accept" recommendation.

Reviewer #4: All comments have been addressed

Reviewer #5: (No Response)

Reviewer #6: All comments have been addressed

2. Is the manuscript technically sound, and do the data support the conclusions?

Reviewer #4: Yes

Reviewer #5: Yes

Reviewer #6: Yes

3. Has the statistical analysis been performed appropriately and rigorously? 

Reviewer #4: Yes

Reviewer #5: N/A

Reviewer #6: Yes

4. Have the authors made all data underlying the findings in their manuscript fully available?

Reviewer #4: Yes

Reviewer #5: Yes

Reviewer #6: Yes

5. Is the manuscript presented in an intelligible fashion and written in standard English?

Reviewer #4: Yes

Reviewer #5: Yes

Reviewer #6: Yes

6. Review Comments to the Author

Reviewer #4: The manuscript is much improved and only very minor things remain. The text can still be shortened in places if you wish.

Table 1: spelling mistake replace with “co-habiting”

Table 4 font of heading too big

Line 355:

If you use the expression “On the other hand” you should use “One the one hand” before and it should describe opposites.

Use when you are comparing two different facts or two opposite ways of thinking about a situation: On the one hand I'd like a job that pays more, but on the other hand I enjoy the work I'm doing at the moment.

Reviewer #5: The authors have answered the reviewers comments. The manuscript is accepted for publication in its present form

Reviewer #6: Table 8: The note at the bottom of the table seems seems redundant. There is no need to mention the significance level of the p value (p<0.05) as it has already been presented in the data analysis section. Here, include only the abbreviations of OR, CI and NI, and list variables included in the multivariate analysis without additional explanation.

7. PLOS authors have the option to publish the peer review history of their article (what does this mean?). If published, this will include your full peer review and any attached files.

Reviewer #4: No

Reviewer #5: No

Reviewer #6: No

---

## [Editor Report · Acceptance letter]

22 Jan 2024

PONE-D-22-34731R2 

PLOS ONE

Dear Dr. Msollo, 

I'm pleased to inform you that your manuscript has been deemed suitable for publication in PLOS ONE. Congratulations! Your manuscript is now being handed over to our production team.

Kind regards, 

on behalf of

Dr. Elingarami Sauli 

Academic Editor

PLOS ONE